# Deep-coverage spatiotemporal proteome of the picoeukaryote *Ostreococcus tauri* reveals differential effects of environmental and endogenous 24-hour rhythms

Holly Kay[1,3], Ellen Grünewald[1,3], Helen K. Feord[1], Sergio Gil[1], Sew Y. Peak-Chew[2], Alessandra Stangherlin [2], John S. O'Neill [2] & Gerben van Ooijen [1✉]

The cellular landscape changes dramatically over the course of a 24 h day. The proteome responds directly to daily environmental cycles and is additionally regulated by the circadian clock. To quantify the relative contribution of diurnal versus circadian regulation, we mapped proteome dynamics under light:dark cycles compared with constant light. Using *Ostreococcus tauri*, a prototypical eukaryotic cell, we achieved 85% coverage, which allowed an unprecedented insight into the identity of proteins that facilitate rhythmic cellular functions. The overlap between diurnally- and circadian-regulated proteins was modest and these proteins exhibited different phases of oscillation between the two conditions. Transcript oscillations were generally poorly predictive of protein oscillations, in which a far lower relative amplitude was observed. We observed coordination between the rhythmic regulation of organelle-encoded proteins with the nuclear-encoded proteins that are targeted to organelles. Rhythmic transmembrane proteins showed a different phase distribution compared with rhythmic soluble proteins, indicating the existence of a circadian regulatory process specific to the biogenesis and/or degradation of membrane proteins. Our observations argue that the cellular spatiotemporal proteome is shaped by a complex interaction between intrinsic and extrinsic regulatory factors through rhythmic regulation at the transcriptional as well as post-transcriptional, translational, and post-translational levels.

[1] School of Biological Sciences, University of Edinburgh, Max Born Crescent, Edinburgh EH9 3BF, UK. [2] MRC Laboratory of Molecular Biology, Francis Crick Avenue, Cambridge CB2 0QH, UK. [3]These authors contributed equally: Holly Kay, Ellen Grünewald. ✉email: Gerben.vanOoijen@ed.ac.uk

Endogenous circadian clocks drive organismal and cellular physiology over the 24 h of the day/night cycle. Before the advent of -omics techniques, reverse genetic approaches identified clock genes in multiple organisms and facilitated the dissection of their auto-regulatory transcriptional–translational feedback loops (TTFLs) in the different taxonomic groups[1]. These feedback loops play a role in the regulation of circadian physiology, metabolism and many other aspects of cellular physiology[2]. Genetic and transcriptomic information are the major source of information upon which models of the cellular clock have been built. However, abundant evidence also suggests that post-transcriptional and post-translational processes are essential to circadian regulation[3–5]. Indeed, in several different contexts, circadian rhythms have been observed in the complete absence of transcriptional feedback[6–9]. Since changes in protein activity underlie every biological process, these studies highlight the importance of studying eukaryotic circadian clocks at the proteome level when investigating the links between environmental signals, TTFLs and post-translational circadian regulation. However, functional circadian proteomics pales in comparison with the detailed characterisation of circadian transcriptomes.

While temporally resolved proteomics datasets exist for a handful of model species (reviewed in ref. [10]), methodological limitations have meant that the coverage of the theoretical maximum proteome is substantially lower than for transcriptomics studies. Lowly abundant proteins, cell cycle proteins, organelle-encoded proteins and transmembrane proteins tend to be underrepresented[11,12], as sample complexity exceeds the detection capacity of mass spectrometric analyses. This is reflected by a maximum of 45% total proteome coverage over circadian time series in the fungus *Neurospora*[12], 30% in *Drosophila*[13], 12% in *Arabidopsis*[14] and 9% in mouse[15]. Furthermore, proteomics studies have exclusively been performed either under rhythmic environmental conditions (e.g. light:dark cycles) or under constant conditions. These experimental approaches are fundamentally different, as they either reflect the combined influences of environmental stimuli and circadian-regulated rhythms, or only the latter. Therefore, a direct comparison between different -omics studies is challenging since experimental details frequently vary[16].

The aim of this study was to provide a detailed analysis of diurnal versus circadian proteomes in a single study, with high proteome coverage including organellar and integral membrane proteins. To reduce sample complexity, we employed the uniquely minimal cellular and genomic complexity of the model cell *Ostreococcus tauri*, a picoeukaryotic alga. This cell type is well established as a cellular model for circadian rhythms across eukaryotes[7,17–19], and is highly amenable to culture under natural diurnal versus constant circadian conditions. Our results provide a rare insight into the complex relationship between environmental and circadian regulation of protein abundance across time, revealing a strikingly differential spatiotemporal proteome under these two conditions.

## Results

**A deep coverage diurnal and circadian proteome**. To attain a deeper understanding of how cellular proteomes change over time, we established extraction procedures to enable increased coverage of transmembrane and organellar proteins in the minimal clock model system *Ostreococcus tauri*[19] (Fig. 1a). We sampled a single day under light–dark entrainment (LD) and three days under constant circadian conditions (LL; Fig. 1b). The longer sampling under constant conditions is necessary to detect repeating patterns and separate true circadian free-running rhythms from noise, and to allow the quantification of

circadian period over three cycles. As cycles are virtually indistinguishable under LD entrainment (refs. [20,21] and Supplementary Fig. 1a), a single cycle was sampled. Based on the negligible difference between biological repeats observed in a pilot experiment (Supplemental Fig. 1b), biological triplicates were pooled to essentially generate a single 'mean' value, before mass spectrometric quantification of the proteome by 11-plex Tandem Mass Tagging Mass Spectrometry (TMT-MS). This methodology allowed the detection of 86% of the 7700 nuclear-encoded proteins, of which 79% were detected at all time points (Fig. 1c and Supplementary Data 1).

To detect ~24 h rhythmicity within the proteome, we compared the methods eJTK[22], RAIN[23] and ECHO[24]. All methods consistently separated rhythmic proteins from arrhythmic proteins under LD (Supplementary Fig. 2a, c). However, a smaller overlap of rhythmic proteins was observed between methods when analysing the LL data (Supplemental Fig. 2b, d). While we provide all results from these three methods in Supplementary Data 2, we will use the eJTK results for all our subsequent analyses because that is the most commonly accepted method in the field and resulted in the more convincing heat maps when applied to our data set. eJTK detected rhythmicity for 67.2% of proteins under LD (Fig. 1d), and 17.9% under LL (Fig. 1e), with a substantial overlap between the two (Fig. 1f). The observation that the *Ostreococcus* proteome is more rhythmic under entrained than constant conditions is consistent with previous observations in mouse[25] and the overall percentage of clock-regulated proteins is within the range observed with other eukaryotes[11–13]. Rhythmic proteins in LL exhibited a mean period of oscillation of 23.3 h (Supplementary Fig. 3a), in line with previous observations with clock luciferase reporter lines under similar conditions[26].

**Verification of the proteome dataset with alternative experiments**. Consistent with previous reports[19,27–29], oscillations in the abundance of the central components of the canonical clock circuit and light perception system were phased similarly under the two conditions (Fig. 2a and Supplementary Fig. 3b–c). We compared the protein profiles of the *Ostreococcus* clock proteins CCA1 and TOC1 with longitudinal imaging of translational fusions of these proteins with firefly luciferase. We observed excellent agreement between luminescent and proteomics-derived traces in terms of phase and amplitude under both conditions (Fig. 2b). As in other species[30], the circadian clock and cell cycle are tightly coupled in *Ostreococcus* and show identical periods[31], leading to the prediction that cell cycle proteins would show similarly organised oscillations between both conditions. We identified 44 core cell cycle candidate proteins based on the *Ostreococcus* genome (Supplementary Data 1). Whilst more of these proteins showed significant daily rhythms under LD than LL, the cell cycle stages inferred from these rhythmically abundant cell cycle proteins are near-identical (Fig. 2a). Our data suggests that the coupling of the cell and circadian cycles is established by a handful of rhythmic regulatory proteins that initiate progression into G1, S, G2 and M phases (Supplementary Fig. 3d–f). As independent verification of these inferred cell cycle stages, we monitored cell division under constant conditions and observed the anticipated relationship (Fig. 2c). Combined, the clock protein luminescence data (Fig. 2b) and matching cell division phases (Fig. 2c) independently validate our proteome dataset as an accurate description of temporal cellular organisation.

**Limited relationship between entrained and free-running rhythms**. The consistent phase of the circadian and cell cycles

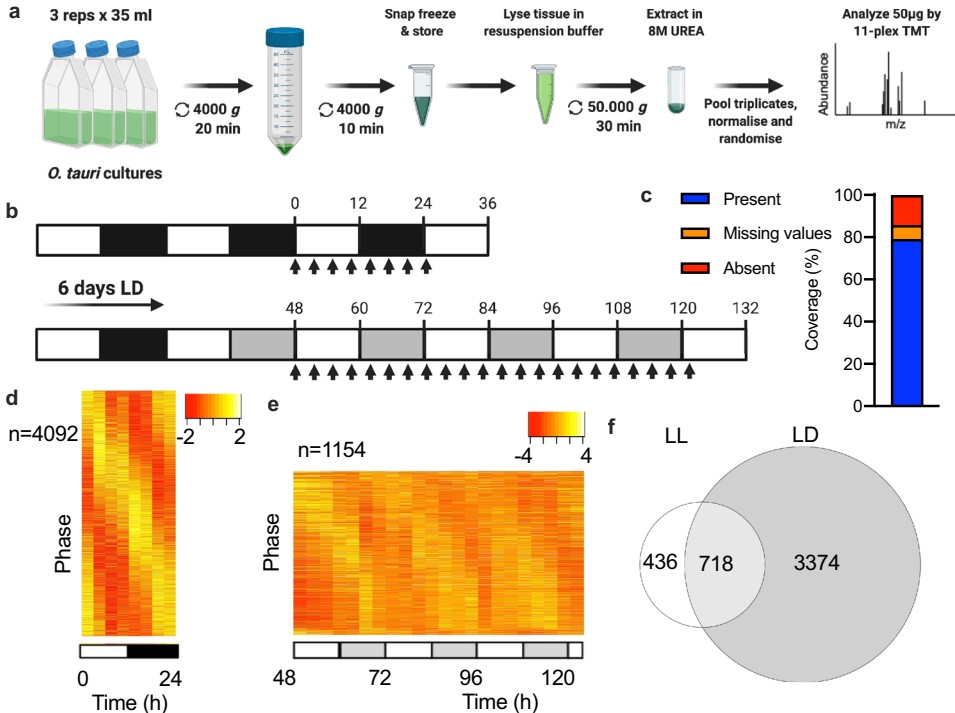

**Fig. 1 Deep-coverage diurnal and circadian proteomes.** An overview of the experimental workflow (**a**) and sampling schedule (**b**) to obtain a deep-coverage proteome. Samples were taken every 3.5 h (arrows) for 1 day under entrained conditions (LD) and for 3 days under constant conditions (LL). **c** The percentage of nuclear-encoded *Ostreococcus* proteins that were quantified at all time points (blue), some time points (orange) or none (red). **d–e** Heat maps showing min–max (red-yellow; Z-scores as indicated) normalised plots for all rhythmic proteins in LD (**d**) or LL (**e**). Rows represent individual proteins, ordered by phase, where each column is a separate time point. **f** Venn diagram showing the overlap between rhythmic proteins under LL and LD, along with rhythmicity percentage under each condition. Source data for **c–f** can be found in Supplementary Data 1.

between LD and LL conditions suggested that clock-regulated output pathways may also be similarly phased. Photosynthesis is one of the most important clock-regulated outputs, but when comparing the rhythmicity parameters of photosynthetic proteins there was little similarity or consistency between the identity or phase of rhythmically abundant proteins under LD compared with LL (Supplementary Fig. 4). While this might be surprising at first sight, the divergence between LD and LL rhythms is clear on the proteome-wide scale. Firstly, while a substantial overlap is observed between rhythmic proteins under LD and LL conditions (Fig. 1f), there is a sizable group of proteins only rhythmic under one of two conditions, indicating a complex interaction between endogenous circadian rhythms with environmental inputs. We observed a significantly higher relative amplitude of oscillations under LD than LL conditions (Fig. 3a). Although the overall difference in means is small, the maximum relative amplitude under LD was 94% versus 38% under LL. Even more notably, the phase distributions of rhythmic proteins were highly differential between LD and LL. The single predominant phase of highest abundance observed under LD was shortly before dawn, while under LL conditions this is in the early subjective night (Fig. 3b). While this difference may be explained in part by the incomplete overlap between proteins rhythmic in LL and LD, the peak abundance phase is distinct even among those proteins that are rhythmically abundant under both conditions (Fig. 3c). As expected, there was no coherence between overrepresented Gene Ontology (GO) or Kyoto Encyclopedia of Genes and Genomes (KEGG) pathways at different phases under both conditions (Supplementary Fig. 5). Together, these data show that the similarly phased cell and circadian cycles under LD versus LL conditions do not lead to coherent functional proteome-wide regulation, and therefore that rhythmicity under entrainment

cannot be inferred from circadian studies under constant conditions alone, or vice versa.

**Limited correlation between transcript and protein abundance rhythms.** Publicly available transcriptomic data in *Ostreococcus* is currently limited to a single microarray study, sampled as three replicate days under entrained LD 12 h light/12 h dark conditions[21]. The gene models used for that study[32] have more recently been updated[33], and upon re-assessment of the probe sequences, we found that 5925 out of 8056 probes map to a unique mRNA. We subjected the microarray data for these probes to the same rhythmicity tests as our proteome dataset and found that nearly all transcripts (98%) exhibited significant rhythms in abundance under diurnal conditions (Fig. 4a), while we found that only 67.2% of proteins are rhythmically abundant under these conditions. The peak phase distribution of transcript rhythms reveals an astonishing bimodal distribution, with peaks in phase around ZT9 and ZT20 (Fig. 4b) just before dusk and before dawn. This previously undetected double wave of transcriptional regulation is consistent with the canonical model of circadian rhythmicity, in which TTFLs allow cells to prepare for the differential demands of day and night through anticipatory changes in gene expression. Implicit to this model is the assumption that mRNA abundance determines protein abundance. However, we observed that the latter shows only a single dawn-phased peak during 24 h (Fig. 3b). This implies that only the pre-dawn peak in gene expression leads to a subsequent peak in protein abundance.

To resolve this apparent paradox, we examined the temporal relationship between those rhythmically abundant proteins that are encoded by rhythmically abundant mRNAs, since it seemed plausible that the transcripts peaking pre-dusk might not encode

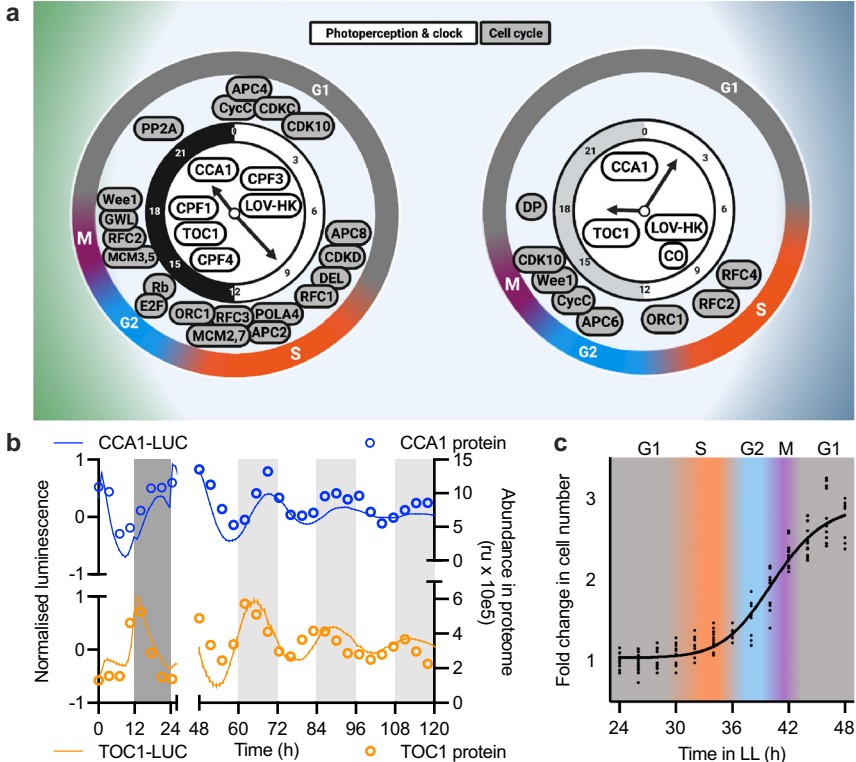

**Fig. 2 Proteome data accurately represent cellular rhythmicity. a** Diagrams depicting the key rhythmically abundant proteins of the circadian and photoperception systems (white) or the cell cycle (grey), expressed on a 24 h clock face based on their peak phase under LD or LL. **b** The relative abundance of TOC1 and CCA1 in LD (0–24 h) and LL (48–120 h) as determined by proteomics (data points, right *Y*-axis) or luciferase results (lines, left *Y*-axis; mean of $n = 30$ (TOC1-LUC) or $n = 48$ (CCA1-LUC)). **c** The cell cycle phases inferred from proteomics, overlaid with observed cell division events under LL. Source data for proteomics results can be found in Supplementary Data 1.

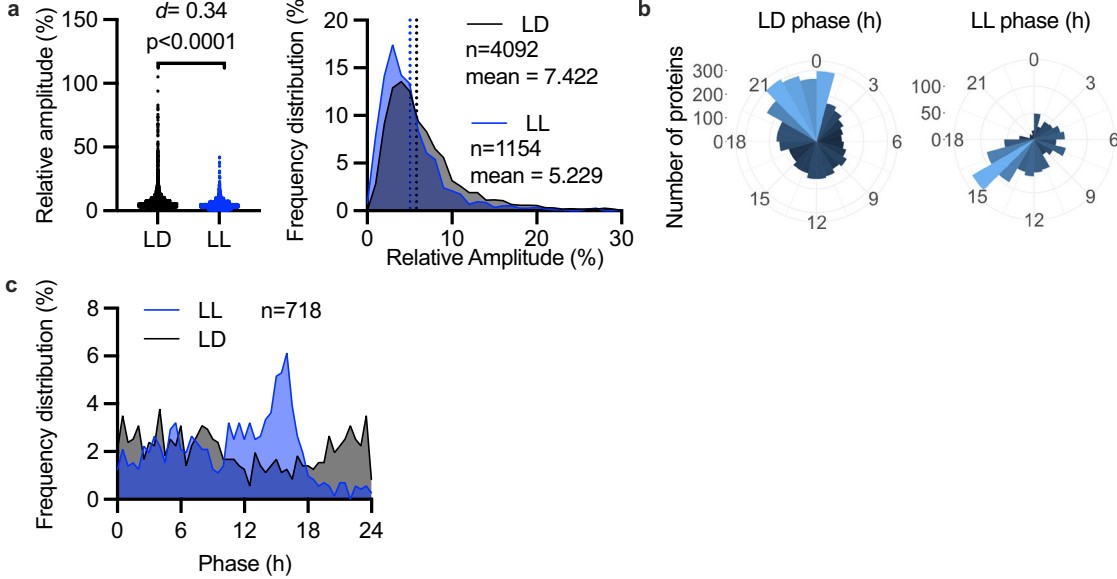

**Fig. 3 No clear relationship exists between entrained and free-running temporal proteomes. a** The relative amplitude of rhythmic proteins under LD or LL, expressed individually or as frequency distributions. **b** Circular histograms showing the number of rhythmic proteins in LD and LL that peak at each 1-hour phase interval. **c** Phase distribution under LD or LL of proteins that were significantly rhythmic in both. Statistics reflect Mann–Whitney tests and Cohen's *d*. Source data can be found in Supplementary Data 1.

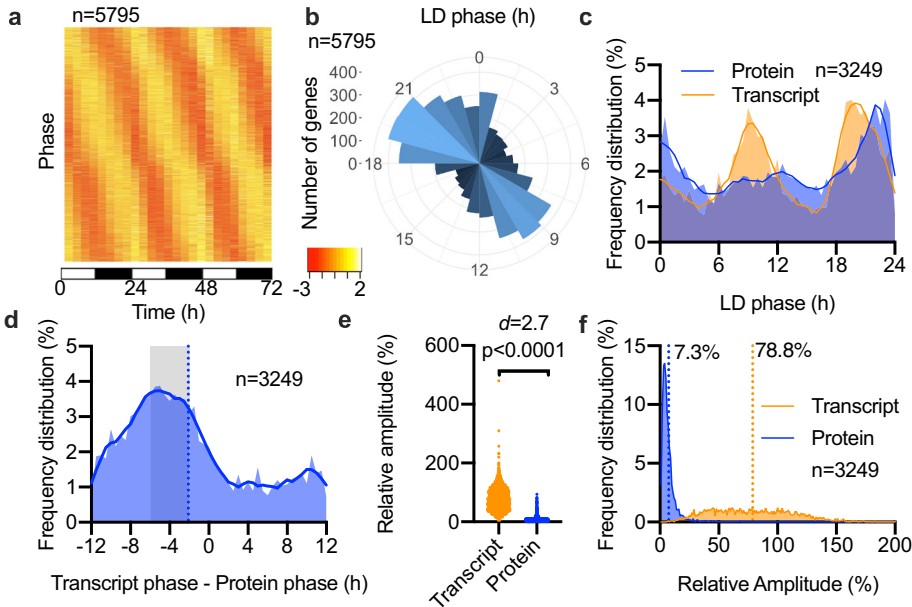

**Fig. 4 Partial correlation between transcript versus protein abundance. a** Heat map showing min–max (red-yellow; Z-scores as indicated) normalised plots for all rhythmic transcripts following a reappraisal of published data[21] under entrained conditions. **b** Circular histogram showing the number of rhythmic transcripts in LD at each 1-hour peak phase interval. **c** Frequency distribution of protein or transcript peak phase under LD conditions, for those where both gene products were rhythmic. **d** One-for-one phase harmonics between rhythmic proteins and their rhythmic transcript, expressed as peak phase of transcript minus peak phase of protein. The mean value is indicated by a dotted line, and values that correlate with a protein peaking in a 2–6 h window after the transcript are indicated with a shaded box. **e–f** Relative amplitude values (**e**) and frequency distribution of relative amplitude (**f**) of transcripts versus proteins under LD conditions. Statistics reflect Mann–Whitney tests and Cohen's d. Source data can be found in Supplementary Data 1.

rhythmic proteins. We identified 3249 such genes among the 4896 genes for which both transcript and protein abundance data are available. Again, we observed 2 daily peaks of transcript abundance but only one for protein abundance (Fig. 4c). In a gene-by-gene comparison of the phase relationship between each transcript with its encoded protein (transcript phase — protein phase), the modal group of proteins peaked 4 h after their transcripts (Fig. 4d). Clear harmony was observed for the protein peak following the transcript peak of the canonical clock protein TOC1 and the photoreceptor LOV-HK, likely involved in entrainment (Supplementary Fig. 6a). These observations are consistent with findings from other organisms[11], and support the canonical model of linear information flow in genetic systems (gene > mRNA > protein). However, only about 30% of the proteins peaked between 2 and 6 h following their cognate transcript, meaning that the majority of proteins did not (Fig. 4d). For example, the transcript and protein peaks of the key cell cycle kinase Wee1 were nearly antiphasic to one another (Supplementary Fig. 6b). This corresponds with observations in other organisms that fewer than half of rhythmic proteins are encoded by rhythmic mRNAs under entrained conditions[25,34]. Finally, transcriptional rhythms showed an average relative amplitude of 78.8%, with transcript levels oscillating by as much as 5-fold (Fig. 4e–f). This contrasts with the mean relative amplitude of 7.3% observed for protein levels under the same conditions, with no proteins exceeding a 1-fold change.

Considering these differences between transcript- and protein-level rhythmicity, we hypothesised that there may be differences in biochemical properties of rhythmic versus arrhythmic proteins that confer a propensity or recalcitrance to circadian regulation. However, no clear differences were observed in protein isoelectric point, predicted disordered regions, hydropathy, protein size, or mean protein abundance under either LD or LL conditions (Supplementary Fig. 7): p values were >0.01 and Cohen's d

statistic values were small, indicating an insignificant proteome-wide effect size. Therefore, it seems unlikely that any of these factors contribute to protein-level rhythmicity. The more parsimonious explanation is that in addition to rhythmic transcriptional information, the light–dark cycle and circadian system affect proteostasis at the post-transcriptional and post-translational levels.

**Differential proteostasis of transmembrane and soluble proteins.** We next compared proteins with predicted transmembrane helices (TM, 1423 proteins) to those without (Soluble, 6276 proteins). A slightly higher proportion of soluble proteins were rhythmic compared with TM proteins under LD (55% vs. 50%) as well as LL conditions (11% vs. 8%). This observation contrasts with comparisons in the mouse liver proteome, in which trans-membrane proteins were more rhythmic than soluble proteins[11]. Conversely, TM proteins were more heavily phosphorylated and more rhythmically phosphorylated than soluble proteins (Supplementary Fig. 8). The highly conserved clock kinases CK1, CK2 and GSK3 are involved in circadian regulation in *Ostreococcus*[7,35,36], yet they were not differentially abundant. These results would correlate with previous studies in mammals showing rhythmic phosphorylation state but constant protein levels for clock-related kinases[37,38].

Interestingly, the peak phase distribution of rhythmic trans-membrane proteins was different from rhythmic soluble proteins: in LD, the phase distribution of TM proteins was bimodal with peaks late in the day and late in the night, whereas soluble proteins peak only at the latter phase (Fig. 5a). Under LL conditions, TM proteins peaked in the middle of the subjective day, whereas soluble proteins peak during the subjective night (Fig. 5b). Peak phases of soluble proteins match the overall proteome peak phases, consistent with soluble proteins dominating the total proteome, while TM protein phases clearly deviate.

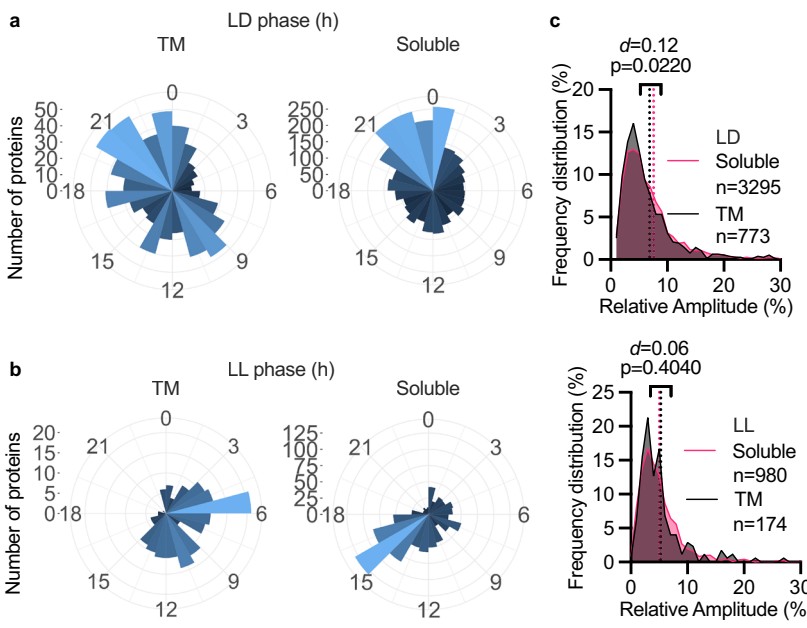

**Fig. 5 Proteins with transmembrane helices are differentially regulated from soluble proteins.** Circular histograms showing the number of rhythmic transmembrane or soluble proteins in LD (**a**) or LL (**b**) at each 1-hour peak phase interval. **c** The frequency distribution of the relative amplitude of transmembrane versus soluble proteins in LD and LL. Statistics reflect Mann–Whitney tests and Cohen's *d*. Source data can be found in Supplementary Data 1.

The mean relative amplitude of rhythmic TM and soluble proteins was not significantly different (Fig. 5c). Together, we conclude that transmembrane proteins are possibly less likely to be rhythmically abundant than soluble proteins, but are more subject to post-translational regulation, which might facilitate circadian regulation of transmembrane transport activity. Additionally, the phase separation between TM and soluble proteins suggests a rhythm in membrane protein biogenesis or degradation that occurs by completely different and previously undetected process to soluble proteins.

**Coordinated regulation of the organellar proteomes.** Following the surprising phase separation of TM versus soluble proteins, we next analysed the organellar proteomes to deepen our understanding of spatial proteome regulation. *Ostreococcus* cells contain a single chloroplast and a single mitochondrion, containing autonomous genomes of 72 and 44 kb, respectively[39]. We detected 79% of the chloroplast-encoded and 63% of the mitochondrial-encoded proteome (Fig. 6a and Supplementary Data 2). Under LD, 48% of the detected chloroplast-encoded proteins were rhythmic (Fig. 6b). The majority of these were ribosomal proteins that peak in the first half of the day, following the two chloroplast-encoded RNA polymerases that peak just before dawn. A smaller number of chloroplast-encoded proteins were rhythmic under LL (Fig. 6b). In contrast, only 26% of mitochondrial-encoded proteins were rhythmic in LD (Fig. 6c), and only two of those were rhythmic in LL (Fig. 6c). This indicates that the circadian system exerts less control over mitochondrial than nuclear or chloroplast gene expression, and that the observed rhythms under entrained conditions are likely to largely result from environmental inputs. Indeed, that would be consistent with the observation that the majority of rhythmic mitochondrial-encoded proteins under entrained conditions peaked around dawn.

The majority of the organellar proteome is made up of nuclear-encoded proteins that are translocated, rather than from organelle-encoded proteins. Therefore, we selected nuclear-encoded proteins carrying a signal peptide targeting it for chloroplast or mitochondrial localisation. The peak abundance phase of nuclear-encoded proteins (solid line in Fig. 6b) with a chloroplast Target Peptide or thylakoid lumen Target Peptide was highly consistent with the phase of chloroplast-encoded proteins (data points in Fig. 6b) under LD as well as LL. Nuclear-encoded sigma factors are required for transcription initiation in the chloroplast and have been shown to confer timing information to the chloroplast in Arabidopsis[40]. We observed a rhythmically abundant sigma factor (SIG6) phased at ZT2 under LD conditions and CT10 under LL conditions that could potentially mediate the environmental and circadian control over chloroplast gene expression in *Ostreococcus* (Supplementary Fig. 9). Nuclear-encoded proteins carrying a mitochondrial Transit Peptide (mTP) peaked around dawn under LD conditions or in the early subjective night under LL conditions (solid lines in Fig. 6c) at highly similar phases as chloroplast-targeted proteins, indicating coordinated regulation of chloroplast and mitochondrial protein abundance.

**Discussion**

We explored the spatiotemporal regulation of a eukaryotic cellular proteome at high depth of coverage. The analysis of such large datasets can be performed in one of several ways, and it is excessively clear that differences in analysis methods can lead to different interpretation of results and therefore different conclusions. We attempted to contribute to constructive debate by assessing protein rhythmicity by three of the main commonly accepted methodologies, and reporting these results and the overlap between them (Supplementary Fig. 2; Supplementary Data 2). While one method is not necessarily better than others, a choice of methods should be based on the requirements and constraints the dataset provides[16], and plotting heat maps of both the rhythmic and the arrhythmic proteins provides an excellent way to visually assess the performance of methodologies on a given dataset. We chose to use eJTK results for our main figures, but this is ultimately a subjective choice, and therefore we provide

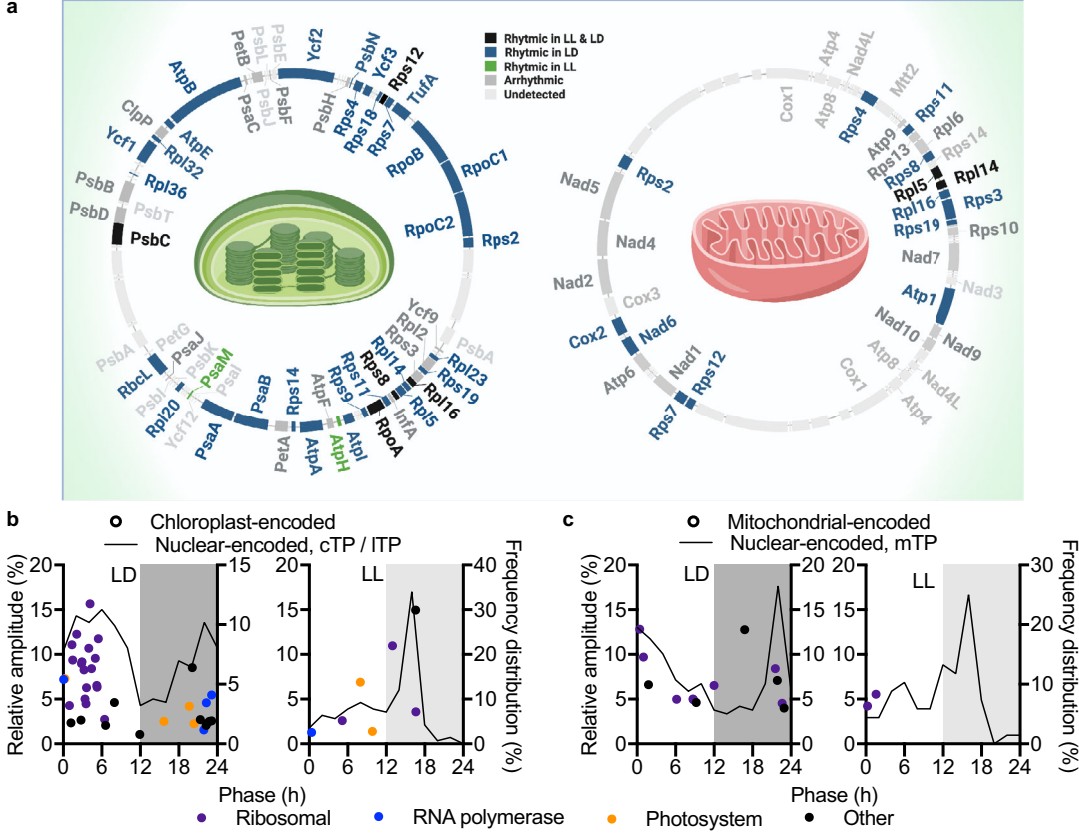

**Fig. 6 Orchestrated rhythmicity of organelle-encoded and organelle-targeted proteomes. a** Overview of the plastid and mitochondrial genome, with rhythmic, arrhythmic and undetected proteins annotated. **b** Phase versus amplitude of rhythmic chloroplast-encoded proteins (data points, left $Y$- axis) under diurnal or constant conditions. Purple dots represent ribosomal proteins, blue RNA polymerases and orange photosystem components. Overlaid is the frequency distribution (black line, right $Y$-axis) of peak phase for rhythmic nuclear-encoded proteins carrying a chloroplast Transit Peptide (cTP) or thylakoid lumen Transit Peptide (lTP). **c** Phase versus amplitude of rhythmic mitochondrial-encoded proteins (data points, left $Y$-axis) under diurnal or constant conditions, overlaid with the frequency distribution (black line, right $Y$-axis) of nuclear-encoded proteins carrying a mitochondrial Transit Peptide (mTP). Source data can be found in Supplementary Data 1.

the full results of other analyses. Would the main conclusions of our paper be different when a different choice was made? Yes and no. The $p$ values of our results would be different using data generated from different methods, which might make some results flip below or above an arbitrary significance threshold. However, when comparisons between large groups of proteins are made (for example rhythmic proteins versus arrhythmic) the numbers involved are exceptionally large, which can lead to statistical significance of differences which in fact have such a small effect size that they are unlikely to be biologically relevant. The effect sizes will not change dramatically between analysis methods (i.e. a large difference is equally observable in the data generated by either of the three methods). For this reason, Cohen's $d$ statistics are provided to accompany all $p$ values ($d$ values above 0.2 are considered small but clear, above 0.5 are medium and above 0.8 large).

Valid conclusions that can be made regardless of analysis method are that complex interactions are observed between clock-regulated rhythms and daily environmental cycles, with a larger number of proteins regulated by the latter. We found little consistency between transcript and protein rhythmicity. Although our proteomics data and the mined transcriptomics data originate from different labs with different growth conditions, the large effect sizes mean that differences in sampling or culturing are unlikely to explain the full divergence between transcript and protein rhythms. We also identified a previously

unknown disconnect between the peak abundance phases of transmembrane and soluble proteins. Conversely, the peak abundance phases of organelle-encoded and organelle-targeted proteins show a high degree of synchrony. Taken together, the key observation that permeated through all our analyses was that no single variable was able to account for protein abundance rhythms. Therefore, rhythmic regulation of protein abundance most likely involves both specific as well as more general mechanisms that together determine the relative balance between protein synthesis versus turnover of each protein.

That conclusion is not controversial, given the large number of processes that exert a documented influence on shaping the spatiotemporal proteome. The rates of transcription, translation, post-translational modification, protein transport and protein degradation are all subject to regulation by transcript-level circadian rhythmicity and by light–dark-dependent regulation[17,41–45]. These considerations necessitate a more inclusive view of what we consider to be the fundamental bases of cellular rhythms. The canonical circadian model suggests that circadian regulation of cell function is driven solely by rhythmic gene expression, established by transcriptional/translational feedback in the core clock. This model assumes linear flow of information: rhythmic regulation of gene expression leads to rhythmic mRNAs, which leads to rhythmic protein levels and rhythmic function. Certainly, the evidence for rhythmic regulation of transcript abundance and the role of clock proteins within

that is irrefutable. However, rhythmic transcription cannot be assumed to elicit comparable changes in protein abundance or activity. Instead, evidence from this study as well as other eukaryotes shows that rhythmicity of protein abundance cannot always be inferred from rhythms in transcript abundance[34,44,46]. This is consistent with a growing number of studies from outside the circadian context that prove that mRNA abundance is poorly predictive of protein abundance or activity[47–49]. In addition, a growing body of evidence for circadian timekeeping without daily cycles of transcriptional activation/repression[50], argues that transcript rhythmicity is not the only physiologically relevant mechanism of timekeeping.

A revised canonical model that might incorporate some of these insights assumes that rhythmic translation of existing mRNA leads to proteome-wide rhythms, which in turn drive rhythmic function. In this model, any mRNA rhythms would be of secondary importance so long as sufficient template is present for each protein's translation. There is sufficient evidence to support that global translation rates are indeed rhythmically regulated[17,51–53]. Our dataset contains a large number of rhythmic ribosomal proteins, which could potentially link transcriptional rhythms to translational rhythms. However, rhythmic translation alone would predict similar peak abundance phases and a far higher overlap between the identity of rhythmic proteins under entrained and constant conditions. Additionally, it cannot explain the different phase of transmembrane versus soluble proteins, nor the coordination of organelle-encoded with organelle-targeted proteins.

A third model for circadian rhythms is that a post-translational oscillator drives rhythmic cell function through rhythmic protein modification and/or degradation rhythms. Here rhythms in transcript abundance result from rhythmic protein activity instead of the other way round, i.e. rhythms in any mRNA or protein abundance are not the cause but a consequence of rhythmic post-translational activities. Prior evidence in *Ostreococcus* for rhythmic regulation in the presence or absence of rhythmic gene expression includes redox-sensitive post-translational modification rhythms and rhythms of $Mg^{2+}$ and $K^+$ transport over the plasma membrane. The observed enrichment for transmembrane transporters among rhythmically phosphorylated proteins likely accounts for these previously reported ion transport rhythms, which is underlined by the low prevalence of transmembrane transport proteins among rhythmic proteins. Integration of our study with the wider current scientific literature provides support for the idea that a combination of transcriptional, translational and post-translational regulation including protein turnover and subcellular trafficking accounts for how ~24 h regulation of eukaryotic cell biology is achieved, similarly to the cyanobacterial circadian system[54]. For example, while the degradation rate of *Ostreococcus* CCA1 protein is rhythmically regulated by the circadian clock, the degradation rate of TOC1 is instead regulated by dark-to-light transitions[43]. Rhythmic protein degradation rates, either of specific proteins or globally, could affect proteome rhythms. Inhibition of proteasome activity delays the Ostreococcus clock at any given phase[43], unlike inhibition of transcription or translation[7]. In mouse liver, protein degradation is highest during the day phase as a result of rhythmic global autophagy, chaperone-mediated autophagy and proteasomal activity[55]. In addition to rates of protein synthesis and turnover, rhythmic localisation within cells is another interesting phenomenon that could influence subcellular protein abundance. We observed a coordinated upregulation of protein abundance among organelle-targeted proteins during the early subjective night. This striking similarity between the peak abundance phases of organelle-encoded and organelle-targeted proteins peak abundance phases indicates a purposeful and coordinated temporal regulation of chloroplast and mitochondrial function. Our cell cycle analysis (Fig. 2c) suggests that this observed peak phase for organellar proteins is consistent with the time of organelle duplication, ahead of cell division later in the night. Clock-dependent coordination of nuclear-encoded, mitochondrial-targeted protein rhythmicity was also observed in mice[56], while nuclear-targeted proteins peak throughout the 24 h cycle[57].

However, the small relative amplitude of rhythmic proteins that we observed (~5%) certainly begs the question of whether protein abundance rhythms are in fact a functionally relevant output of transcriptional rhythms at all. It seems more likely that transcript rhythmicity presents a means of achieving proteostasis, to counteract a rhythmic requirement driven by rhythmic function and associated rhythmic protein turnover. This interpretation would be supported by the recent observation in mammalian cells that more rhythmic proteins are observed without a transcriptional core clock than are observed with one[58]. Perhaps rhythmic function generates rhythmic gene expression as a consequence of rhythmic post-translational regulation and protein degradation, leading to a back to front transcriptional/translational clock model. Ultimately, to fully understand eukaryotic cellular timekeeping, a complete picture of circadian post-transcriptional, translational and post-translational regulation is needed alongside transcriptional regulation. To obtain comprehensive models of the cellular circadian landscape, it will be necessary to integrate different levels of organisation by multi-omics approaches. This integration relies on high-quality datasets, and by contributing the highest-coverage proteome data over full diurnal and circadian time series available across Eukaryota, we have provided a step in this direction.

## Methods

**Sample collection and preparation**. All cell lines (WT, CCA1-LUC and TOC1-LUC) are *Ostreococcus tauri* OTTH0595 (RCC745) and were obtained from the Bouget lab (Laboratoire d'Océanographie Microbienne). *Ostreococcus tauri* cells were cultured in artificial seawater (ASW) and entrained under cycles of 12 h light:12 h dark (LD) as reported previously[17] for 6 days. Twenty-four hours prior to the first sampling point, cultures were either transferred to constant light (LL) or kept in LD cycles (LD, Fig. 1a). Samples were collected in triplicate every 3.5 h for 3 full cycles in LL (22 time points) and one full cycle in LD (8 time points). For each time point whole cells were collected at 4000 rpm for 20 min. Media was discarded and each cell pellet was gently resuspended in 0.9 ml ASW. Cells were collected at 4000 rpm for 10 min and the media discarded. After adding two chrome beads (3 mm) to each pellet the samples were snap-frozen in liquid $N_2$. Samples were stored at −80 °C until all time points were collected. Resuspension buffer (50 mM Hepes, pH 7.5, 150 mM NaCl, protease inhibitors (Roche)) was added to each frozen pellet on ice. Cells were lysed using a Tissue Lyser (Eppendorf) in precooled blocks (1 min at 30/s). Whole lysates were centrifuged at 50,000 g for 30 min at 4 °C in a Beckman Optima MAX ultracentrifuge. Pellets were kept on ice and were washed once carefully with resuspension buffer. Pellets were resuspended in 8 M urea buffer (8 M urea, 20 mM Tris-HCl, pH 8) by vortexing. Samples were then sonicated in a Bioruptor (Diagenode) for 30 s on/30 s off (x5). All samples were centrifuged at 17,000 g for 10 min to remove unsolubilised debris. Triplicate samples per time point were pooled, samples were randomised and 50 μg of each time point was analysed by 3 sets of 11-plex TMT.

**TMT peptide labelling**. Samples were randomised before allocating to TMT runs, and the operator was blinded to the sample IDs. Samples were trypsin-digested as reported previously[58]. Lyophilised peptides were resuspended in 20 μl of 175 mM triethylammonium bicarbonate and labelled with a distinct TMT tag, 12 μl, from a stock prepared as per manufacturer's instructions (Thermo Scientific), for 60 min at room temperature. The labelling reaction was quenched by incubation with 2.2 μl 5% hydroxylamine for 30 min. Labelled peptides from 10 time point samples and 1 pool were combined into a single sample and partially dried to remove MeCN in a SpeedVac (Savant). Samples were desalted and the eluted peptides were lyophilised.

**Basic pH reverse-phase HPLC fractionation and LC–MS/MS**. The TMT labelled peptides were subjected to off-line High-Performance Liquid Chromatography (HPLC) fractionation[58]. The fractionated peptides were analysed by LC–MS/MS using a fully automated Ultimate 3000 RSLC nano System (Thermo Scientific) fitted with a

100 μm × 2 cm PepMap100 C18 nano trap column and a 75 μm × 25 cm reverse-phase NanoEase M/Z HSSC18 T3 column (Waters). Samples were separated using a binary gradient consisting of buffer A (2% MeCN, 0.1% formic acid) and buffer B (80% MeCN, 0.1% formic acid), and eluted at 300 nL/min with an acetonitrile gradient. The outlet of the nano column was directly interfaced via a nanospray ion source to a Q Exactive Plus mass spectrometer (Thermo Scientific). The mass spectrometer was operated in standard data-dependent mode, performing a MS full-scan in the m/z range of 380–1600, with a resolution of 70,000. This was followed by MS2 acquisitions of the 15 most intense ions with a resolution of 35,000 and Normalised Collision Energy (NCE) of 33%. MS target values of 3e6 and MS2 target values of 1e5 were used. The isolation window of precursor ion was set at 0.7 Da and sequenced peptides were excluded for 40 s.

**Spectral processing and protein identification**. Raw files were processed using MaxQuant[59] v 1.6.6.0. MS/MS spectra were quantified with reporter ion MS2 and searched against the nuclear-encoded proteome obtained from the Ostta V2.2 database[33,60] plus the mitochondrial and chloroplastic genomes[39]. Carbamidomethylation of cysteines was set as fixed modification, while methionine oxidation, N-terminal acetylation and phosphorylation of serine, threonine and tyrosine, were set as variable modifications. Protein quantification requirements were set at 1 unique and razor peptide. In the identification tab, second peptides and match between runs were not selected. Other parameters in MaxQuant were set to default values. The MaxQuant output file was processed with Perseus (v1.6.6.0). Identifications from the reverse database were removed, only identified by site, potential contaminants were removed, and we only considered proteins with ≥1 unique and razor peptide. All columns with an intensity "less or equal to zero" were converted to "NAN" and exported as a.txt file. The MaxQuant output file with phosphor (STY) sites table was also processed with Perseus software (v1.6.6.0). The data was filtered: identifications from the reverse database were removed, potential contaminants were removed and we only considered phosphopeptides with localisation probability ≥0.75. Then all columns with intensity "less or equal to zero" were converted to "NAN" and exported as .txt file.

**Normalisation**. Since an equal amount of protein was used for each TMT labelling reaction, sample loading normalisation was performed by taking the sum of all intensities for each time point, and normalising to the mean of these. Internal reference scaling (IRS)[61] was then carried out to allow for comparisons between TMT experiments: the mean abundance for each protein in each of the three pools was calculated. The mean of these means was calculated and used to normalise the value for each protein between the three TMT runs. In instances where a protein was missing from a pool sample, the mean of the remaining 2 pool samples was used to normalise. For all except five proteins in the dataset, if a protein was missing from the pool sample it was also missing from every individual time point in that TMT run. Peptides that are not unique to one protein were removed, as were proteins that were detected at too low levels to reliably quantify. Time points were then de-randomised to obtain the final data set.

**Circadian parameter estimations**. The JTK_cycle algorithm with empirical calculation of p-values (eJTK)[22] was performed using BioDare2[62]. All time points were included in the rhythmicity analysis, with linear detrending of input data for LL and no detrending for LD, and parameters as preset for 'eJTK Classic'. The cut-off for rhythmicity was at $p < 0.05$, as customary for eJTK. The RAIN method[23] was implemented using the rain() function in the R/Bioconductor software package. For LD data, 'period' was set to '24', 'period.delta' was '0' and 'method' was 'independent'. For LL, 'period' was set to '24', 'period.delta' was '4' and 'method' was 'longitudinal'. For both conditions, 'deltat' was '3.5' and 'nr.series' was '1'. All other parameters were left as default. The cut-off for rhythmicity was at $p < 0.05$, as customary for RAIN. ECHO was implemented using the echo.find() function in the ECHO R package[24]. For LD data, both 'low' and 'high' were set to '24' and 'is_de_linear_trend' was left as 'FALSE'. For LL, 'low' was set to '20' and 'high' was set to 28, while 'is_de_linear_trend' was 'TRUE'. For both conditions, 'begin' and 'end' were set to the corresponding time points for the time series, 'resol' was '3.5' and 'num_reps' was '1'. All other parameters were left as default. The cut-off for rhythmicity was at $p < 0.05$, as customary for ECHO.

For all rhythmic proteins, BioDare2[62] (biodare2.ed.ac.uk) was used to calculate all circadian parameters. The MFourFit algorithm was used for absolute phase and amplitude calculation with amplitude and baseline detrending. Data were $log_2$ transformed prior to all rhythmicity and parameter calculations, except for amplitude values. LD data were concatenated to be able to approximate these parameters, and the period was constrained to 23.5–24.5 h. The MESA algorithm with amplitude and baseline detrending was used for period calculation of the LL data, constrained to 18–34 h. To transform absolute phase to circadian phase, absolute phase predictions by MFourFit were divided by 24, and multiplied by the period as estimated by MESA. Proteins with any missing values in the LD dataset were omitted from the analysis but kept in the data set with circadian parameters as Not Determined (ND). For the LL dataset, a maximum of 1/3rd of missing values was allowed, equating to those missing values for one out of three TMT runs. Phosphosite data were normalised as described above. For LL, phosphosites

detected in two-thirds of time points were kept in the dataset, and for LD only those present in all time points were retained. Prior to rhythmicity and circadian parameter analyses, the phosphosite data were normalised to their protein abundance across the time points. Phosphosites for which the protein had not been detected in the dataset were removed. Rhythmicity and circadian analyses were carried out as above.

**Biological verification experiments**. Raw luminescence data for Fig. 2 was detrended by subtracting a rolling average of luminescence readings for the following 24 h[26]. For cell proliferation analyses, cultures subjected to the identical conditions as described for the proteomic analyses were sampled every 2 h on the second day of constant light. Cells were counted under a light microscope using a haemocytometer. Two biological replicates were performed, with 5 technical replicates for each time point.

**Transcript analysis**. The probe sequences from a publicly available O. tauri microarray study under entrained LD conditions[21] were originally designed using outdated gene models and were therefore blasted against the O. tauri genome V2.2 on the Orcae service[33,60]. This resulted in usable microarray data for 5925 of the 7700 genes in the genome. Circadian parameters were estimated as outlined for the proteome dataset above, using eJTK without detrending for rhythmicity analysis and MFourFit for phase and amplitude calculation, with period constrained to 23.5–24.5 h.

**Structural and functional protein data**. Gene ontology and KEGG pathway analyses were performed in R v3.6.1. The enrichGO and enrichKEGG functions from the clusterprofiler R package[63] were used to compare a target dataset to background. Transmembrane helices in proteins were predicted using web tool TMHMM server 2.0[64]. Proteins with at least 1 predicted transmembrane helix were considered 'transmembrane' in subsequent analyses. The Sequence Manipulation Suite web tool was used to calculate protein molecular weight and isoelectric point[65]. N-terminal presequences in the entire nuclear-encoded proteome were identified with TargetP-2.0[66]. Those containing a predicted mitochondrial transit peptide (mTP) were used to generate a 'mitochondrial proteome', and those containing either a chloroplast transit peptide (cTP) or thylakoid luminal transit peptide (luTP) were used for the 'chloroplast proteome'. Hydrophobicity and intrinsic disorder of proteins were calculated by localCIDER package[67] in Python 3.x, with the get_kappa and get_uversky_hydropathy functions used for intrinsic disorder and hydrophobicity, respectively[68,69]. A low kappa value implies a propensity to form random coils and therefore higher intrinsic disorder. Radial plots were created using the ggplot2 R package. Venn diagrams were generated using the eulerr R package. To generate heat maps, the rhythmic proteins/transcripts were ordered by their calculated phase (absolute phase for LD proteins/transcript or phase in circadian time for LL proteins). The abundance of each protein was normalised by the time course mean of the protein, and values were centred around 0 using the scale function in R before applying the heatmap.2 function from the pvclust R package[70]. Relevant proteins from different biological processes were depicted in diagrams based on the following cell categories: cell cycle[21,31], light signalling and clock[29,71,72], photosynthesis and chloroplast biosynthesis[73], and organelle-encoded proteins[39]. Bioinformatic analyses to identify potential candidate proteins were conducted using Standard Protein Basic Local Alignment Search Tool (BLAST) of known protein sequences from other model systems as Arabidopsis thaliana and Homo sapiens using the Ostreococcus genome ORCAE V2[60]. Graphs, density plots and histograms were plotted and statistics were calculated using GraphPad Prism v8, R v3.6.1. Figure 1a and Supplementary Fig. 3b, d–e, 4a–d were made with BioRender.

**Statistics and reproducibility**. Where data followed normal distribution, we used two-tailed t tests. Where data did not follow normal distribution, we used Mann–Whitney U two-tailed tests. Cohen's d was calculated for all. Sample sizes are reported on the figure in each case, and refer to numbers of proteins/genes (the full proteomics experiment was performed once).

**Reporting summary**. Further information on research design is available in the Nature Research Reporting Summary linked to this article.

## Data availability

The mass spectrometry proteomics data have been deposited to the ProteomeXchange Consortium via the PRIDE[74] partner repository with the dataset identifier PXD025009. Supplementary Data 1 contains the source data for all main figures except Fig. 2c, and for Supplementary Figs. 3, 4, 6, 7, 8a–b, and 9. Supplementary Data 2 contains source data for Supplementary Fig. 2.

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

## Acknowledgements
The authors would like to acknowledge Jennifer Hurley, Jackie Pelham, Andrés Romanowski, Babette Vlieger, Francisco José Romero-Campero, Ana Belén Romero Losada and Phil Kirk for bioinformatics support. E.G. was supported by a Wellcome Trust Institutional Strategic Support Fund award to G.v.O., H.K. by a Royal Society Research Fellows Enhancement Award to GvO (RGF\EA\180192) and S.G. by a Leverhulme Trust Research Grant awarded to GvO (RPG-2019-184). H.K.F. is a PhD student funded by the Biotechnology and Biological Sciences Research Council (BBSRC, BB/M010996/1). A.S. was supported by the AstraZeneca Blue Skies Initiative. J.S.O. was supported by UKRI Medical Research Council (MC_UP_1201/4). GvO was supported by a Royal Society University Research Fellowship (UF110173) and renewal (UF160685).

## Author contributions
GvO and JSO conceived the approach. GvO provided the overall supervision and management of the project. The protocol development and all pilot studies were carried out by EG. Time series sampling and processing of samples were performed by EG and HF. AS contributed to developing the detection strategy using TMT labelling. S.P.C. labelled the samples and performed mass spectrometric detection. Bioinformatic analysis of the resultant data and generation of figures was carried out by HK, SG and GvO. All authors interpreted the results. HK, SG and HF wrote the first draft. GvO wrote the final draft on which all authors commented.

## Competing interests
The authors declare no competing interests.
