## [Transparent Peer Review File · Communications Biology]

Reviewers' comments:

Reviewer #1 (Remarks to the Author):

This study by Kay and Grunewald and colleagues provides an in-depth circadian examination of the proteome in the eukaryotic green algae *Ostreococcus*. The widespread use of new techniques for genomics has rapidly increased our knowledge of circadian transcriptomics, while our understanding of the changes at the protein level have lagged behind. In this interesting study, the investigators compared the temporal dynamics of the proteome under both constant light conditions and under light-dark conditions. The consistency of the period length of the temporal proteome under constant conditions with previous research provides validation of the methods. Unexpectedly, the authors found comparatively little overlap (11%) between the diurnal proteome and the circadian proteome, with core clock proteins and cell cycle proteins demonstrating the greatest similarity. These interesting results will undoubtedly intrigue circadian researchers. This paper presents a lot of interesting results that are contained in the Supplemental material that perhaps should be included as main figures. It makes it harder for the reader for key results to be in supplemental figures. While it is known from other studies that rhythmic RNA abundance does not guarantee rhythms in protein abundance, it is unclear as to whether the microarray data set the authors use provide enough evidence for circadian transcriptomics. The experiments are well-designed. The manuscript is well-written making it accessible for a broad readership.

Comments:

1. The reanalysis of the previous microarray data set may be over interpreted as the microarray data does not provide an unbiased data set for RNA abundance. In their results section, the authors should provide brief details of the microarray data set and the link to the publicly available data as this is the first time it is mentioned.
2. The authors need to provide more rationale and explanation of the analysis in the Properties of rhythmic versus arrhythmic proteins. While I appreciate the attempt to consider many factors, there should be explanation as to what factors could influence the protein rhythmicity. Figure 4 may be better suited as supplemental.
3. Figure 4 and its analysis need more explanation. It is unclear as to whether total protein abundance was examined or individual protein abundance under the two conditions was examined. Were all time points and samples pooled for this analysis? It seems as if this
4. Hydrophobicity - section Differential properties of proteins. In the first sentence under this section, this is the first time this result is mentioned. The previous reference to the Supplemental figures 5A and B referred to the isoelectric point of the proteins and the disordered nature of some proteins. This observation needs to be more detailed. Given the reliance of the hydrophobicity data from Supplemental Figure 5, it would perhaps be more suited to include this as part of the Main figure 5.
5. Given the importance of post-translational modifications both in clock function and regulation of outputs, it seems the analysis of the phosphorylation of proteins would receive more than one or two sentences in the results and in supplemental figure 6. Again, this information may be more suited as part of a main figure.
6. Given the amount of information in this paper, the Discussion seems on the brief side. I think there needs to be a more in-depth discussion of the results.

Reviewer #2 (Remarks to the Author):

This article by Kay and colleagues describes the high coverage temporal proteomic analysis of *Ostreococcus tauri* under entrained (light/dark cycle) and circadian (constant light) conditions. It leads to the observation that only a small number of proteins are rhythmic under circadian condition compared to entrained conditions. As described in mammals, in both conditions, amplitudes are very

low and there is a very limited correlation between mRNA and protein levels. This constitutes a well performed and analysed experiment and a well written manuscript. Only a few minor revisions are necessary:

- One problem with the analysis resides in the fact that mRNA expression has been measured in another lab, meaning different experimental conditions. This could participate in the discrepancy between mRNA and proteins rhythms, despite a few controls with clock proteins. This problem is indicated in the manuscript but need to be more emphasized, particularly in the discussion.
- Authors did not discuss a potential role of protein degradation in the generation of protein rhythms, as described in mammals (PMID: 30759397).
- The description of the potential role of protein transport is also limited despite evidence in mammals (PMID: 26862173, 27818260).
- Some references do not fit with the associated sentences. For example, reference 15 is cited for: proteome coverage in Drosophila, rhythm in constant conditions, rhythmic proteins and mRNA correlation and regulation by light in mouse liver. This reference is about rhythmic mRNA degradation under light/dark cycle! Same problem for ref 47: this article did not describe the role of light! Other references need to be cited! Same issue for rhythmic mRNA translation: some references are more appropriate than 44 (e.g. PMID: 23300384, 26486724, 26554015, 26338483, 27506798)

Reviewer #3 (Remarks to the Author):

The manuscript "Differential effects of environmental and endogenous 24h rhythms within a deep-coverage spatiotemporal proteome" investigates the relationship between the proteins that oscillate in constant conditions, and are therefore controlled by the circadian clock, and those that oscillate under diurnal conditions, and there for integrate both circadian and diurnal cues to time these oscillations. The authors show that there is a significant difference between oscillations in constant and in diurnal conditions and conclude that there must therefore be, as has been predicted by many other proteomics papers, a significant level of post-transcriptional regulation in the clock. While the effort of the authors is significant and presents potential insights into clock timing of output, there is a major flaw in the acquisition and analysis of the data that needs to be fixed before conclusions can be drawn from the work. In addition, the major conclusion that is drawn by the authors, that the clock is first a protein clock, is not well supported by the data that they have, and the authors would need to revise the conclusion extensively, or more work would need to be done to support the conclusion. I address my two major criticisms below and withhold many of the minor criticisms of the paper because the reanalysis that would be necessary to improve this work would likely lead to changing some of these minor issues.

Major comments:

1) Why did the authors decide to use only 24 hours of data for the entrained condition while using 72 hours of data for the constant condition? This has led to several problems.

a) It is clear in the constant condition that there is a strong dampening of the signal, using luciferase traces and their heat maps as an example. This is an issue because rhythms that dampen could be missed because eJTK cannot handle damping of the signal.

b) Moreover, having only 24 hours of data for the entrained condition makes it much easier to find false positives in the entrained condition vs the constant conditions.

c) The authors did not use the same method to analyze the two different data sets. This differential analysis could have led to potentially large artifacts in the data.

d) Also, why did the authors not run their biological replicates independently, which could have increased the confidence in their identified rhythms?

All of the above might account for some of the 4X increase in diurnal conditions and certainly would change the downstream evaluation of the data. The best option to fix this problem would be to gather the later time points for the diurnal samples and make a full comparison. If this is not possible, the authors need to either reanalyze the data using programs that can handle both types of data sets and

damping of the signal by either detrending or model creation (Metacycle, ECHO, Rain) or they need to cut off the subsequent 48 hours of constant condition data and leave only the first 24 hours so that they are comparing apples to apples. Without this major change in the analysis, it is hard to lend credence to even the most basic conclusions of the author.

2) In their conclusion section, the authors surmise that because there is a great deal of post-transcriptional regulation of circadian output, the basic model of the clock where transcription is the driver of time-keeping is no longer plausible. However, their conclusion is not supported by the data that they present. There are many other plausible explanations as to where this post-transcriptional regulation is coming from, including clock interactions with ribosomal proteins and transcriptional oscillations in U^{BQ} levels (both suggested in the circadian literature). However, this data does not support or refute any of the alternative models of clock regulation, it only shows that there is more complexity in the circadian timekeeping loop than is currently accounted for.

Minor comments:

1) In figure 1H, it is not clear what the y axis parameter Cell proliferation means.

2) At times, the authors define something as significant that appears only to be so because of the large numbers of proteins analyzed, which the authors note in the text. A Cohen's D statistic would help the authors to assess the significance of the effect (e.g. S-fig 5, Fig2).

3) The authors speak often in hyperbole, sometimes beyond what is true. E.g., their last statement includes "by contributing the most detailed temporally resolved proteome across Eukaryota" which is not true, as there are many more detailed proteomics works that have been completed.

Reply to reviewers

Dear reviewers,

First of all, a big thank you for the effort you have all put in to improve our manuscript. All your comments have made this version of the manuscript so much better.

Before addressing all your comments individually, I would like to make some general comments on the new version of the manuscript. We believe that the best way to establish constructive debate about our results is through openness and clarity on why and how we came to the analyses we did. In light of the reviewers' comments and the wider discussion in the field, we now include two new supplemental figures and one additional supplemental table relating to the data analysis (as detailed in the reply to each reviewer below). While creating these resources, we have indeed identified a problem with the original analyses of our LL data that led to underreporting of rhythmic proteins under LL. We are thankful for the functioning of the review process (in this regard especially reviewer 3), and relieved to have caught this and other weaknesses in our paper.

We hope that the reviewers can all agree that the current version of the manuscript reports an accurate description of the data we obtained, and that it contributes to open and honest discussion in our community in a constructive manner.

Below, reviewers' comments are quoted in italic font.

On behalf of all authors,
Gerben van Ooijen,

Reviewer 1

"1. The reanalysis of the previous microarray data set may be over interpreted as the microarray data does not provide an unbiased data set for RNA abundance. In their results section, the authors should provide brief details of the microarray data set and the link to the publicly available data as this is the first time it is mentioned."

It is true that transcriptomic data is generally lacking in this organism. All that is currently available is a single microarray study under LD12:12. There are no published transcriptomics data under free-running conditions. As this original study used incorrect gene models, not all transcripts were represented in the data. We have simply re-analysed all original microarray probes, which leads to a coverage of 74% of the full transcriptome. The 'bias' in that dataset therefore originates from whether the gene was correctly modelled in the original genome version. Rhythmicity parameters in this LD transcriptome were analysed identically to the rhythmicity parameters of our LD proteome to be able to compare between the two, showing that 98% of transcripts are rhythmic. The heat map clearly indicates the validity of that claim, as it is beautifully rhythmic. The full documentation of this re-analysis is available within supplemental file 2. As you suggested, we have expanded the brief description of the original experimental design in the main manuscript (page 4 "Publicly available transcriptomic data in *Ostreococcus* is currently limited to a single microarray study, sampled as three replicate days under entrained LD 12h light / 12h dark conditions", and made an extra disclaimer in the discussion section (page 6 "We found little consistency between transcript and protein rhythmicity. Although our proteomics data and the mined transcriptomics data originate from different labs with different growth conditions, the large effect sizes mean that differences in sampling or culturing are unlikely to explain the full divergence between transcript and protein rhythms").

"2. The authors need to provide more rationale and explanation of the analysis in the Properties of rhythmic versus arrhythmic proteins. While I appreciate the attempt to consider many factors, there should be explanation as to what factors could influence the protein rhythmicity. Figure 4 may be better suited as supplemental."

We agree, and all these results can now be found together in supplemental figure 7. Our aim here was to explicitly NOT analyse only factors for which we could clearly see how it could affect rhythmicity. If transcript rhythmicity does not equate linearly to protein rhythmicity, this exercise was intended to find any other previously unknown explanations. We did not find much evidence to suggest there are obvious differences, so it is mainly a negative result and we agree it is better placed in a supplemental figure. The description in the results section has been streamlined (page 4).

“3. Figure 4 and its analysis need more explanation. It is unclear as to whether total protein abundance was examined or individual protein abundance under the two conditions was examined. Were all time points and samples pooled for this analysis? It seems as if this “

Original figure 4A (now supplemental figure 7c) shows the mean abundance of each individual protein across LD timepoints versus the mean abundance of it across all LL timepoints; either for the rhythmic proteins (blue), arrhythmic proteins (orange) or all proteins (dotted line). These data are taken from the columns labelled ‘mean abundance LD’ and ‘mean abundance LL’ in supplemental data file 1. It should be clearer in the current wording.

“4. Hydrophobicity - section Differential properties of proteins. In the first sentence under this section, this is the first time this result is mentioned. The previous reference to the Supplemental figures 5A and B referred to the isoelectric point of the proteins and the disordered nature of some proteins. This observation needs to be more detailed. Given the reliance of the hydrophobicity data from Supplemental Figure 5, it would perhaps be more suited to include this as part of the Main figure 5.”

Thank you for this suggestion. Following our re-analysis of the dataset it is now clear that the effect size of this observation is so small that it is not likely to be biologically significant (even though the statistical p value is <0.05). We have made a version of our manuscript with that result in the main figure, but given the simplification of our narrative in this section, it ultimately makes more sense to combine this result with the other results in supplemental figure 7. We have a new section in the discussion that describes our view that differences with tiny effect sizes, statistically significant or not, are unlikely to be biologically meaningful (page 6).

“5. Given the importance of post-translational modifications both in clock function and regulation of outputs, it seems the analysis of the phosphorylation of proteins would receive more than one or two sentences in the results and in supplemental figure 6. Again, this information may be more suited as part of a main figure.”

We understand the logic of this, and indeed this result was part of a main figure in a previous iteration of the manuscript. However, we feel in the end that the phospho-peptide dataset is too small to make a really big point out of it: we did not perform phospho enrichment to obtain these data, but simply report on phosphopeptides we happened to detect in the whole proteome. Alternative studies exist that have better coverage of phosphopeptides, and in the end we thought it would be better to cite those studies and put our new data in a supporting supplemental file (rather than not reporting them at all, which would seem wasteful). So respectfully, we do think it is better to keep it there. We explicitly state in the legend to supplemental figure 8 that this is not from phospho-enriched samples.

“6. Given the amount of information in this paper, the Discussion seems on the brief side. I think there needs to be a more in-depth discussion of the results.”

The new manuscript indeed contains a longer discussion section, including new sections on data analysis methods and on the role of protein degradation and transport in proteostasis. Overall, the discussion has expanded from 983 to 1403 words while the results section has been simplified, so we hope you find it more balanced now.

Reviewer 2

“One problem with the analysis resides in the fact that mRNA expression has been measured in another lab, meaning different experimental conditions. This could participate in the discrepancy between mRNA and proteins rhythms, despite a few controls with clock proteins. This problem is indicated in the manuscript but need to be more emphasized, particularly in the discussion.”

Yes, this is a fair point. We have included more detail about the original microarray study in the main result, and added a sentence to the discussion to make this point (page 6 “We found little consistency between transcript and protein rhythmicity. Although our proteomics data and the mined transcriptomics data originate from different labs with different growth conditions, the large effect sizes mean that differences in sampling or culturing are unlikely to explain the full divergence between transcript and protein rhythms”).

“-Authors did not discuss a potential role of protein degradation in the generation of protein rhythms, as described in mammals (PMID: 30759397).

and

- *The description of the potential role of protein transport is also limited despite evidence in mammals (PMID: 26862173, 27818260).*”

Of course protein degradation and transport are major contributors to proteostasis, and we have now expanded our discussion around post-translational rhythms (page 7) to include those things. The suggested references were very useful, thank you.

“- Some references do not fit with the associated sentences. For example, reference 15 is cited for: proteome coverage in *Drosophila*, rhythm in constant conditions, rhythmic proteins and mRNA correlation and regulation by light in mouse liver. This reference is about rhythmic mRNA degradation under light/dark cycle! Same problem for ref 47: this article did not describe the role of light! Other references need to be cited! Same issue for rhythmic mRNA translation: some references are more appropriate than 44 (e.g. PMID: 23300384, 26486724, 26554015, 26338483, 27506798)”

We apologise for these errors, and we have made the required changes. Thank you for bringing this to our attention.

Reviewer 3

“1) Why did the authors decide to use only 24 hours of data for the entrained condition while using 72 hours of data for the constant condition? This has led to several problems.”

and

“d) Also, why did the authors not run their biological replicates independently, which could have increased the confidence in their identified rhythms?”

I will combine my reply to these two points as the answer overlaps. Obviously the LL data need to run longer than a single cycle, because to assess circadian parameters such as period you need that replication in your dataset. This is also critical to tell truly circadian rhythms (that persists with a FRP of ~24h) from non-rhythmic proteins; you need to be able to see the repeating pattern. Under entrainment, you simply don't have this necessity: period is 24h by definition, and expression or abundance patterns from entrained conditions are remarkably identical between days. For example, look at figure two of Troein et al., 2011 (PMID 21219507) where four days luminescent data of *Ostreococcus* transcriptional AND translational markers of both CCA1 and TOC1 are presented: all four days are near-identical. It is not contentious for diurnal timeseries to cover a single cycle, especially in costly -omics studies, and often the biological replicates are plotted sequentially to visualise sequential patterns. For example, the microarray study from *Ostreococcus* that we use was sampled as three replicates of a single day, with overlapping timepoints at ZT0, and these three days were stitched together in the original publication to make repeating patterns more visible (Monnier et al., 2010: PMID 20307298). Sampling one day under LD conditions makes perfect sense, in a way that it does not for LL conditions. We have made that point clear in the result section now. We also included a new supplemental figure (Supplemental Figure 1a) to show how similar, qualitatively, clock gene expression is under LD. Finally, we added references to the above publications to make our point clear: page 3 “We sampled a single day under light-dark entrainment (LD) and three days under constant circadian conditions (LL; Fig. 1B). The longer sampling under constant conditions is necessary to detect repeating patterns and separate true circadian free-running rhythms from noise, and to allow the quantification of circadian period over three cycles. As cycles are virtually indistinguishable under LD entrainment (20,21 and Supp. Fig. 1A), a single cycle was sampled.”

Why then did we not run our biological replicates separately? Firstly, that decision was taken after extensive experimentation ahead of our time series analysis. In pilot experiments we did perform separate biological replicates. We have prepared a new supplemental figure (Supp Fig 1B) to indicate this effect: we took 10 proteins at random from our pilot data, covering a range of overall protein abundance, and show that the results are incredibly reproducible. It became abundantly clear that the error bars were insignificant and did not warrant the tripling of costs, processing time, and facility running times. In fact, our decision to still sample three biological replicates and pool these prior to MS analysis (to essentially generate ‘mean only, n=3’ data) was born out of an abundance of caution. We make clear statements in the legend to supp fig 1b as well as the main text (“Based on the negligible difference between biological repeats observed in a pilot experiment (Supplemental Fig. 1B), biological triplicates were pooled to essentially generate a single ‘mean’ value, before mass spectrometric quantification of the proteome by 11-plex Tandem Mass Tagging Mass Spectrometry (TMT-MS).”.

Let me describe why we have practically no detectable variation between replicates, and equally, why we would not expect to see differences between multiple days under entrainment. We are working with a highly uniform clonal cell line that is neither adherent nor actively motile, but freely moves around by Brownian motion (yes, that's how small the cells are). This is a very different type of biological sample than for example a set of *Arabidopsis* plants where two plants are never identical, or a number of mouse livers that have to be surgically

removed from different animals, or a collection of *Drosophila* heads that are separated from their bodies by vortexing a heterogeneous group of whole animals. We simply do not require the same level of processing: we spin down identical volumes (containing 30 million individuals per mL) of three identical and highly homogeneous cultures, and freeze the pellets. Yes, biological replicates are often included when using other organisms, but these are taken on the same day in the same experiment rather than in truly independent experiments, and therefore merely indicate the variation between biological material and extract preparation, neither of which are a source of variation in this cell type. A statement to that extent is now also included in the new legend to Supp Fig 1.

"a) It is clear in the constant condition that there is a strong dampening of the signal, using luciferase traces and their heat maps as an example. This is an issue because rhythms that dampen could be missed because eJTK cannot handle dampening of the signal. "

b) Moreover, having only 24 hours of data for the entrained condition makes it much easier to find false positives in the entrained condition vs the constant conditions."

c) The authors did not use the same method to analyze the two different data sets. This differential analysis could have led to potentially large artifacts in the data."

"All of the above might account for some of the 4X increase in diurnal conditions and certainly would change the downstream evaluation of the data. The best option to fix this problem would be to gather the later time points for the diurnal samples and make a full comparison. If this is not possible, the authors need to either reanalyze the data using programs that can handle both types of data sets and dampening of the signal by either detrending or model creation (Metacycle, ECHO, Rain) or they need to cut off the subsequent 48 hours of constant condition data and leave only the first 24 hours so that they are comparing apples to apples. Without this major change in the analysis, it is hard to lend credence to even the most basic conclusions of the author. "

I will reply to these points together, because a description of our new analyses will address all these points. First of all, an important note is that we have realised that by mistake the LL data was analysed without the detrending option that eJTK provides. This detrending is important to remove the effect of dampening that is observed under constant conditions. The reviewer was totally right that this led to an underreporting of rhythms in LL: rhythmic proteins reported by eJTK in LL have gone up from 11% to 18% as a result, and the heat maps clearly indicate that this is an improvement.

We also totally concede that we did not make the most optimal choices in our initial analysis, and have scrapped the cosinor analysis from the manuscript entirely. We now analysed LD and LL data using the same methods: eJTK, RAIN and ECHO, and include the results for all these analyses in the new supplemental figure 2 and new supplemental file 2. In Supp fig 2, we plot the resultant heat maps of 'rhythmic' and 'arrhythmic' proteins as called by the three different methods, because the better rhythmicity predictions will produce the more striking rhythmic pattern in a heatmap for 'rhythmic' proteins, and a lack of repeating patterns in a heat maps of the 'arrhythmic' proteins. The overlap between the three methods applied to our LD data is very extensive, as can be seen from the Venn diagram in Supp Fig 2C. The fact that the 'rhythmic' heat maps show fantastic wave-like abundance rhythms phased throughout the day, whereas the 'arrhythmic' groups do not, indicates that our LD analysis is not overreporting rhythms, nor identifies false positives. For the LL data, the overlap between methods is still substantial and the percentage rhythmicity across methods is similar too. However, we believe that the eJTK analysis leads to the more appropriate 'rhythmic' and 'arrhythmic' heat maps. We therefore used the results of the eJTK analysis in our main figures. However, we have made this caveat explicit in the first section of results (page 3), and added a whole new paragraph about it to start the discussion section (page 6).

Finally, just a simple reminder that the values that we get for percentage of rhythmic proteins under entrainment or in free running conditions are not dissimilar to what has been found in other eukaryotes, as we cited in the original rebuttal. Our results are not outlandish in that respect.

Overall, the reviewer's comments under point 1) have been absolutely instrumental to our improved analysis and presentation of the data, and have led directly to two new supplemental figures, a supplemental table, and substantial rewriting of the results and discussion section. It has made us realise a flaw in our original analysis, so it's been great. I do hope you find this new version more palatable.

"2) In their conclusion section, the authors surmise that because there is a great deal of post-transcriptional regulation of circadian output, the basic model of the clock where transcription is the driver of time-keeping is no longer plausible. However, their conclusion is not supported by the data that they present. "

With all respect, the reviewer's description of our conclusion, above, is not accurate. We did state that transcriptional rhythms have a clear and undeniable effect in shaping proteome rhythms, but that they are not sufficient to account for all aspects of rhythmicity. This conclusion in fact IS supported by our data in the context of the wider scientific literature. Of course, in a discussion section you contextualise the results of the current study with what is already out there. We propose that additional layers of regulation exist, that of course

interact with transcriptional rhythms but are not necessarily transcription-driven (as evidenced by rhythms in the absence of transcription). Given the changes we made to the discussion section, including a strong effort to tone down our enthusiasm, we hope that the phrasing is now more acceptable to the reviewer.

“There are many other plausible explanations as to where this post-transcriptional regulation is coming from, including clock interactions with ribosomal proteins and transcriptional oscillations in UBQ levels (both suggested in the circadian literature). However, this data does not support or refute any of the alternative models of clock regulation, it only shows that there is more complexity in the circadian timekeeping loop than is currently accounted for.”

The conclusion the reviewer has just reached is indeed closer to our own conclusion. It is interesting that the reviewer cites these two specific examples; indeed we have noticed extensive abundance rhythms in ribosomal proteins, and indeed we have previously shown how important targeted degradation is for the clock (in general, but regulated degradation of TTFL components too). We have added a statement around rhythmic regulation of ribosomes to the relevant discussion section (“Our dataset contains a large number of rhythmic ribosomal proteins, which could potentially link transcriptional rhythms to translational rhythms.”), and given the importance of degradation to proteostasis, we have included a new paragraph in the discussion section (see also reply to rev2).

“Minor comments:

1) In figure 1H, it is not clear what the y axis parameter Cell proliferation means.”

Yes I totally agree, and we have changed this y-axis (‘Fold change in cell number’).

“2) At times, the authors define something as significant that appears only to be so because of the large numbers of proteins analyzed, which the authors note in the text. A cohen’s D statistic would help the authors to assess the significance of the effect (e.g. S-fig 5, Fig2).”

Thank you for the suggestion, we have now indeed used Cohen’s d statistics. It has been very helpful to support the point we had indeed made in the original version more clearly.

“3) The authors speak often in hyperbole, sometimes beyond what is true. E.g., their last statement includes “by contributing the most detailed temporally resolved proteome across Eukaryota” which is not true, as there are many more detailed proteomics works that have been completed. “

It would have been helpful if the reviewer added PMIDs to this, because we have searched far and wide, and are quite sure our statement is accurate. More likely perhaps; could it be that our phrasing is confusing? What we mean by the statement is that there is no time series study (circadian or diurnal) available, in any eukaryotic organism or cell type, with a coverage near what we achieved. Even though we believe the original statement is accurate, we have revised it so it should no longer be open to other interpretations: “To obtain comprehensive models of the cellular circadian landscape, it will be necessary to integrate different levels of organisation by multi-omics approaches. This integration relies on high-quality datasets, and by contributing the highest-coverage proteome data over full diurnal and circadian time series available across Eukaryota, we have provided a step in this direction”.

REVIEWERS' COMMENTS:

Reviewer #1 (Remarks to the Author):

In the revised manuscript by Kay, Grunewald and colleagues, the authors have answered many concerns from the original manuscript with a more thorough discussion and clarity provided on the limitations of the study. However, it seems as if many of the interesting features of the study have now been relegated to preliminary and supplemental information such as the protein phosphorylation or perhaps not of physiological significance such as the protein hydrophobicity results. Consequently, the study seems to rely even more heavily on the differences between the single published microarray study and the current proteomics study in LD and LL conditions. Although I think this is a serious limitation of the work, I agree that the authors' study provides a useful resource and an important step for future studies

Reviewer #2 (Remarks to the Author):

The authors have adequately addressed reviewers' queries.

Reviewer #3 (Remarks to the Author):

In the revision of the manuscript "Differential effects of environmental and endogenous 24h rhythms within a deep-coverage spatiotemporal proteome" the authors have clearly spent a great deal of time addressing the issues that I highlighted. Their restructuring of the discussion section is appreciated and fits what I was hoping for. They have also spent a great deal of time reanalyzing their data in the results section, which has led to a better understanding of what is and is not oscillating in these cells.